# Antibiotic Exposure Concurrently with Anti-PD1 Blockade Therapy Reduces Overall Survival in Patients with Child–Pugh Class A Advanced Hepatocellular Carcinoma

**DOI:** 10.3390/cancers16010133

**Published:** 2023-12-27

**Authors:** Kanan Alshammari, Faizah M. Alotaibi, Futoon Alsugheir, Mohammad Aldawoud, Ashwaq Alolayan, Mohammed Ahmad Algarni, Fouad Sabatin, Mohammad F. Mohammad, Abdulaziz Alosaimi, Faisal M. Sanai, Hassan Odah, Ahmed Saleh Alshehri, Omar S. Aldibasi, Samah Alrehaily, Abdullah S. Al Saleh

**Affiliations:** 1King Abdulaziz Medical City, Ministry of National Guard-Health Affairs, Riyadh 11481, Saudi Arabia; alshammarika@mngha.med.sa (K.A.); futoonhihs@gmail.com (F.A.); mfdriot@hotmail.com (M.A.); alolayanas@mngha.med.sa (A.A.); garnimo2@gmail.com (M.A.A.); sabatinfo@mngha.med.sa (F.S.); alosaimiab@ngha.med.sa (A.A.); aldibasiom@ngha.med.sa (O.S.A.); alsalehab1@mngha.med.sa (A.S.A.S.); 2College of Medicine, King Saud Bin Abdulaziz University for Health Sciences, Riyadh 11481, Saudi Arabia; 3King Abdullah International Medical Research Center, Ministry of National Guard-Health Affairs, Riyadh 11481, Saudi Arabia; 4College of Science and Health Professions, King Saud Bin Abdulaziz University for Health Sciences, Alahsa 31982, Saudi Arabia; 5Abdominal Imaging Section, Department of Radiology, King Faisal Specialist Hospital & Research Center, Riyadh 11564, Saudi Arabia; 6King Abdulaziz Medical City, Ministry of National Guard-Health Affairs, Jeddah 21423, Saudi Arabia; sanaifa@mngha.med.sa (F.M.S.); dr.hassan-@hotmail.com (H.O.); aalshehri.md@gmail.com (A.S.A.); samah.alrehaily@yahoo.com (S.A.); 7College of Medicine, King Saud Bin Abdulaziz University for Health Sciences, Jeddah 21423, Saudi Arabia; 8King Abdullah International Medical Research Center, Ministry of National Guard-Health Affairs, Jeddah 21423, Saudi Arabia

**Keywords:** antibiotic, Nivolumab, immune checkpoint inhibitor (ICI), anti-PD-1, hepatocellular carcinoma (HCC)

## Abstract

**Simple Summary:**

This study examines the role of antibiotic use on the immunotherapeutic response in patients with hepatocellular carcinoma (HCC) treated with an immune checkpoint inhibitor (ICI). HCC is the leading cause of cancer death worldwide. In this work, we examine 59 patients with HCC treated with ICI, 39 patients did not use antibiotics, and 20 patients did use antibiotics concurrent with ICI. We found that patients with Child–Pugh class A advanced HCC who did not take antibiotics during the treatment course had significantly longer overall survival compared to those who did. This work adds to the growing evidence that antibiotic use during the treatment course with ICI might negatively affect survival outcomes. This work along with others suggests the need for a careful prescription of antibiotics to patients with HCC undergoing ICI.

**Abstract:**

Hepatocellular carcinoma (HCC) is the third leading cause of cancer death worldwide with a poor prognosis. Treatment with immune checkpoint inhibitors (ICIs) has improved overall survival in patients with HCC. However, not all patients benefit from the treatment. In this study, 59 patients with HCC were enrolled from two medical centers in Saudi Arabia, with 34% using antibiotics concurrently with their Nivolumab (anti-PD1 blockade). The impact of antibiotic use on the clinical outcomes of patients with HCC undergoing treatment with anti-PD1 blockade was examined. The patients’ overall survival (OS) was 5 months (95% CI: 3.2, 6.7) compared to 10 months (95% CI: 0, 22.2) (*p* = 0.08). Notably, patients with Child–Pugh A cirrhosis receiving anti-PD1 blockade treatment without concurrent antibiotic use showed a significantly longer median OS reaching 22 months (95% CI: 6.5, 37.4) compared to those who were given antibiotics with a median OS of 6 months (95% CI: 2.7, 9.2) (*p* = 0.02). This difference in overall survival was particularly found in Child–Pugh class A patients receiving anti-PD1 blockade. These findings suggest that antibiotic use may negatively affect survival outcomes in HCC patients undergoing anti-PD1 blockade, potentially due to antibiotic-induced alterations to the gut microbiome impacting the anti-PD1 blockade response. This study suggests the need for careful consideration when prescribing antibiotics to patients with HCC receiving anti-PD1 blockade.

## 1. Introduction

Human microbiota consists of trillions of microorganisms that reside in our bodies and have been recognized for their impact on health and disease [1]. Many of these microorganisms reside in the gut, creating a community known as the gut microbiota [2]. These microorganisms contribute to different functions ranging from synthesizing enzymes and vitamins as well as aiding the digestive system [2]. Furthermore, the influence of the gut microbiota expands to impact the immune system and the immune cells [3]. It plays a critical role in shaping and modulating the immune system via different mechanisms, including stimulating the differentiation of the immune cells and the production of anti-inflammatory subsets. Dysbiosis, which is characterized by the disruption of the diversity or composition of the gut microbiota, has been associated with the initiation of various diseases, impacting treatment outcomes in diseases like cancer [4]. In the previous few years, many studies have shifted toward the role of gut microbiota in cancer immunotherapy outcomes, particularly immune checkpoint inhibitors (ICIs) [5,6,7]. ICIs have shown promising results for the treatment of different types of cancer, including melanoma, non-small cell lung cancer, and hepatocellular carcinoma (HCC) [8,9]. It works by blocking inhibitory signaling on immune cells and enhancing the immune response against cancer cells. For example, Nivolumab targets programmed death 1 (PD-1), which, by binding to its ligands, programmed death ligand 1 (PD-L1) and programmed death ligand 2 (PD-L2), can inhibit T cell activation. Similarly, Ipilimumab targets the cytotoxic T-lymphocyte-associated antigen 4 (CTLA-4) on T cells and blocks its interaction to inhibit the costimulatory signals [10]. HCC is the leading cause of cancer-related death worldwide [11] and is often diagnosed at an advanced stage and, therefore, has limited treatment options [12]. Treatment with ICIs has shown promising results in improving overall survival compared to tyrosine kinase inhibitors in advanced stages [13]. This was shown in the IMbrave150 and HIMALAYA trials, which reported an overall survival benefit of combination therapies (of PDL-1 inhibitors combined with anti-VEGF or CTLA-4 inhibitors) over sorafenib [14,15]. These studies resulted in FDA approvals of Atezolizumab and Bevacizumab, and more recently Durvalumab and Tremelimumab for the treatment of unresectable HCC. Approved second-line ICI therapies after progression on sorafenib include Nivolumab with or without Ipilimumab, and Pembrolizumab [16,17]. However, not all patients respond to ICI treatment, and the selection of suitable candidates who are likely to benefit from ICI treatment is a complex process that requires careful consideration of different factors, including the type of cancer, stage, patient comorbidities, anti-drug antibodies [18], and the molecular characteristics of the tumor [19].

Research suggests that the gut microbiota may contribute to ICI treatment responses, potentially due to their role in immune activation and inflammation [20]. The use of antibiotics can significantly alter the gut microbiota’s composition and diversity. While antibiotics are indispensable in treating bacterial infections, their indiscriminate use can lead to dysbiosis, potentially affecting the gut microbiota’s beneficial roles [21,22]. Emerging evidence suggests that antibiotic use can impact the efficacy of ICI therapy, likely due to its effects on the gut microbiota [23,24]. Understanding this relationship is crucial as it could influence the selection criteria for ICI therapy candidates and inform strategies to mitigate any potential negative impacts of antibiotics.

In this study, we evaluated the medical records of 59 patients with HCC who received ICI with or without the use of antibiotics. We explored the association between antibiotic use and the clinical outcomes of ICI in patients with HCC. 

## 2. Materials and Methods

### 2.1. Study Population

This study was conducted on a cohort of 59 patients diagnosed with advanced HCC, who were selected from two academic centers in Saudi Arabia. A majority of these patients progressed on sorafenib treatment and subsequently received Nivolumab as part of their therapeutic regimen in the second-line setting, while 13.6% and 3.4% received Nivolumab as a first- and third-line setting, respectively. The cohort was chosen to represent a specific patient population with advanced HCC and Child–Pugh classes A and B who received Nivolumab as a systemic therapy. Inclusion criteria included adult patients with advanced HCC—BCLC C or multifocal BCLC B not amenable for locoregional therapy, with Child–Pugh A, or early 7B cirrhosis. Excluded patients were ones with BCLC D HCC, Child–Pugh C cirrhosis, presence of brain metastases, known HIV/AIDS, or presence of concurrent malignancy (other than known HCC) and Eastern Cooperative Oncology Group (ECOG) ≥ 3. Patients with underlying autoimmune liver disease, advanced chronic kidney disease (stage ≥ 4), and cholestasis (bilirubin > 34 µmol/L) were also excluded.

### 2.2. Variables and Outcomes

The primary variables under investigation included patient characteristics (such as age, gender, and clinical history), tumor data (including stage and histology), and antibiotic use. Antibiotic administration was specifically tracked for up to 30 days prior to the initiation of Nivolumab therapy, during the treatment course, and up to 30 days post-therapy. The patients’ cancer staging was determined using the BCLC staging system and the severity of cirrhosis was assessed using Child–Pugh scores.

The primary outcome of interest was the overall survival (OS) time. The OS was calculated from the date of initial Nivolumab treatment to the date of death or last follow-up. Secondary outcomes included the response rate to Nivolumab therapy (partial or complete response), and the impact of Child–Pugh score on survival outcomes.

### 2.3. Statistical Analysis

The cohort was divided based on the use of antibiotics, and Child–Pugh class. The baseline characteristics of the cohorts were compared using either Chi-squared or Fisher’s exact test as appropriate. The overall survival was analyzed using the Kaplan–Meier method, and the log-rank test was used to compare the survival curves between the group without antibiotics and the group with antibiotics based on the overall cohort and the Child-Pugh class. The level of significance was set at *p* < 0.05 for all statistical tests. All analyses were performed using SPSS statistical software version 29.0.1.0 (New York, NY, USA). 

## 3. Results


**Patient characteristics:**


A total of 59 patients with HCC were included in this study. Table 1 shows the baseline characteristics based on antibiotic use (39 patients did not use antibiotics, and 20 patients used antibiotics). The median age of the cohort was 72 (65, 79) years, 70 years (62, 78) for the group without antibiotics, and 73 (57, 81) for the group with antibiotics with males representing the majority in both groups at 89.7% and 80%, in the group without antibiotic use and the group with antibiotics, respectively. The mean body mass index (BMI) was 26.5 for the group without antibiotics and 24.5 for the group with antibiotics. 

With regard to performance status, patients were classified based on their ECOG scores, with 66.7% scoring 0–1 and 33.3% scoring 2–3 in the group without antibiotic use, while 80% scored 0–1 and 20% scored 2–3 in the group with antibiotics. A total of 64.1% of patients were classified as Child–Pugh A, and a total of 35.9% were classified as Child–Pugh B in the group without antibiotics. Similarly, in the group with antibiotics, 65% and 35% were classified as Child–Pugh A and Child–Pugh B, respectively. In the group without antibiotics, the cancer staging was 28.2% and 71.8% at stages B and C, respectively, and in the group with antibiotics, the cancer staging was 30% and 70% of patients at stages B and C, respectively (Table 1). 

The primary etiology of cirrhosis in the group without antibiotics was hepatitis B (30%) and hepatitis C (15.4%), and in the group with antibiotics was hepatitis B (20%) and hepatitis C (30%) (Table 1). Furthermore, the most common comorbidities observed among the patients were cardiac disease (15.4% and 25%), diabetes mellitus (56.4% and 50%), chronic kidney disease (7.7% and 10%), hypertension (38.5% and 55%), cirrhosis (87.2% and 80%) in the group without antibiotics and the group with antibiotics, respectively (Table 1). Furthermore, a subgroup analysis of the group with antibiotics is shown in Table 2. All patients received antibiotics within 30 days of Nivolumab administration (pre- or post-Nivolumab). The majority of the group received Ceftriaxone to treat spontaneous bacterial peritonitis (SBP) (30%) or urinary tract infection (UTI) (5%). Ciprofloxacin was the second-highest antibiotic used to treat UTI (25%). Moreover, Piperacillin/tazobactam and Meropenem were used to treat sepsis, 15% and 5%, respectively. A similar percentage of the patients received Augmentin (10%) to treat upper respiratory tract infections (URTIs) and Moxifloxacin (10%) to treat pneumonia.


**Treatment interventions:**


The treatment interventions are summarized in Table 3. The majority of patients in both the group without antibiotics and the group with antibiotics received Nivolumab as a second line of therapy, 84.6% vs. 80%, respectively. Nivolumab, as a first-line therapy, was given to 15.4% of patients with no antibiotic use and to 10% of the patients with antibiotic use (*p* = 0.12). None of the patients in the no-antibiotic-use group received Nivolumab as a third-line therapy, while 10% of patients in the antibiotic-use group received Nivolumab as a third-line therapy. Both groups had a similar number of therapy cycles 7 vs. 6.5, in the group without antibiotics and the group with antibiotics, respectively. Other classes of used medications were not different between the two groups (*p* > 0.05 in all). During their treatment course, beta-blockers were given to 28.2% of patients without antibiotic use and to 40% of patients with antibiotic use. Oral hypoglycemics were used among 23.1% of patients without antibiotic use and 10% of patients with antibiotic use. Furthermore, proton pump inhibitors were given to 17.9% of patients with no antibiotic use and to 20% of patients with antibiotic use. A similar percentage of patients in the group without antibiotics and in the group with antibiotics were given steroids at 35.9% vs. 30%, respectively (Table 3). 


**Treatment response and survival curves**


We further examined patients’ survival curves based on both Child–Pugh class A and B and found no significant differences between the group with antibiotics vs. the group without antibiotics (*p* = 0.08; Figure 1A). Notably, the median overall survival in patients with Child–Pugh class A significantly decreased (*p* = 0.02; Figure 1B) in patients with antibiotic use (6 months, 95% CI: 2.7, 9.2) compared to patients without antibiotic use (22 months, 95% CI: 6.5, 37.4). Patients’ survival curves based on Child–Pugh class B (Figure 1C) were similar between both groups (*p* = 0.48; Figure 1C). 

The treatment responses in patients are described in Table 4. The modified response evaluation criteria in solid tumors (mRECIST) were used to assess the treatment response in patients who did not use antibiotics during ICI therapy (group without antibiotics) and in patients who used antibiotics during the therapy (group with antibiotics). 

In mRECIST (Table 4), a total of 41 patients were assessed since some patients had clinical decompensation that precluded them from receiving further systemic therapy, and hence no disease assessment scans were done subsequently. Only one patient achieved complete response (CR) in the group with antibiotics compared to none in the group without antibiotics. A similar percentage number of patients in the group without antibiotics and the group with antibiotics achieved partial response (PR), 40% vs. 42.9%, respectively. The group without antibiotics was 18.5% progressed disease (PD) compared to 14.3% in the group with antibiotics. With regard to stable disease (SD), the group without antibiotics and the group with antibiotics achieved similar results, 37% vs. 42.9%, respectively. 

Multivariate Cox regression analysis identified only antibiotic use as affecting the overall survival independently from other co-variants, including the age and gender as well as the risk of co-morbidities such as history of diabetes mellitus, hypertension, cardiac disease, chronic kidney disease, and presence of cirrhosis (Table 5). 

## 4. Discussion

Our study provides evidence showing the concurrent use of antibiotics during treatment with ICI can have a significant detrimental impact on the overall survival in patients with HCC and Child–Pugh class A cirrhosis, which is the recommended class to receive ICIs by the FDA. ICI has shown promising results in improving overall survival in several cancer types, including HCC, which remains one of the leading causes of cancer-related deaths worldwide [11]. However, not all patients benefit from the treatment, and our study, along with others [25,26,27,28,29], suggests that the use of antibiotics concurrently with ICI could contribute to reduced overall survival. 

Our findings show a significant association between antibiotic use and reduced overall survival in patients with HCC and Child–Pugh class A cirrhosis undergoing ICI treatment. This finding aligns with growing evidence suggesting the gut microbiota, which can be influenced by Ceftriaxone [30], Ciprofloxacin [31], Pipracillin/tazobactam [32], Meropenem [33], Augmentin [34], and Moxifloxacin [35] antibiotics used in our cohort, as indicated in Table 2, significantly play a crucial role in regulating response to ICIs [23,36]. Furthermore, no differences in the overall survival in patients with HCC and Child–Pugh class B cirrhosis undergoing ICI treatment. This could be due to the disease severity in patients with Child–Pugh class B that could affect overall survival and eliminate any differences that may occur due to changes in the gut microbiota that occurred after antibiotic uses. 

The gut microbiota consists of millions of diverse microorganisms that show a critical influence on our health and disease [1]. Several studies have reported an important relationship between the gut microbiota and the host immune cells and its crucial role in reshaping and modeling the immune response to ICI treatment. The use of antibiotics can disturb the gut microbiota as it has a wide range of impacts on different types of microorganisms. This disruption in the gut microbiota is known as dysbiosis [4], which could potentially impact the response to ICI and thereby explain the reduced overall survival in patients with HCC who received antibiotics concurrently with their ICI treatment.

Our findings emphasize the need for a more careful approach to antibiotic prescription in HCC patients receiving ICIs. The potential implications of antibiotic-induced alterations to the gut microbiota on ICI treatment outcomes should be thoroughly considered when making clinical decisions. However, given the indispensable role of antibiotics in managing infections, striking a balance between their necessary use and the potential adverse effects on ICI efficacy will undoubtedly pose a considerable challenge.

Further studies are important to fully understand the complex relationship between antibiotics, ICIs, and clinical outcomes. In addition, explore potential strategies to eliminate the negative effects of antibiotics on ICI treatment. This could involve the development of more targeted antibiotics that preserve the beneficial gut microbes or the use of probiotics and other microbiota-modulating interventions to restore the gut microbiota after antibiotic treatment.

### Limitations

Despite the statistically significant finding of reduced median overall survival in patients with Child–Pugh class A who used antibiotics vs. those who did not, our study has a few limitations. Approximately one-third of our patients had Child–Pugh class B cirrhosis, and these also have significantly worse overall survival compared with Child–Pugh class A patients. Furthermore, our study did not show differences in the partial or complete responses to ICIs between patients who received antibiotics and those who did not, according to the mRECIST disease response criteria. Instead, the impact became evident over a longer period of time. However, the exact timeline of these effects is not clear. This distinction warrants further investigation to determine the precise mechanisms through which antibiotics may affect long-term survival without significantly altering the initial responses to ICIs. We also did not record locoregional therapy administration prior to systemic therapy use in BCLC B patients, and this may have an impact on survival. Another bias would be that patients who are prescribed antibiotics are mostly ones with some degree of infection/sepsis, and a worse overall state of health, which on its own may negatively impact their survival. Lastly, our relatively low number of patients in this study may have underestimated any real differences in the radiological tumor responses, which were similar between the two groups. Furthermore, it is important for future studies to highlight the role of infections, which are the major cause for using antibiotics as a factor in impairing overall survival and response to ICI.

## 5. Conclusions

Our study adds to the growing body of evidence suggesting that antibiotic use may negatively affect survival outcomes in HCC patients undergoing ICI treatment. These findings highlight the crucial role of the gut microbiome in modulating the response to ICIs and emphasize the need for an alternative approach to antibiotic use in this patient population. As we continue to unravel the complex interactions between the gut microbiota and the immune system, these insights may pave the way for more personalized and effective therapeutic strategies in the fight against HCC. Our study suggests a link between antibiotic use and treatment outcome but does not negate the potential role of the infection in reducing overall survival.

## Figures and Tables

**Figure 1 cancers-16-00133-f001:**
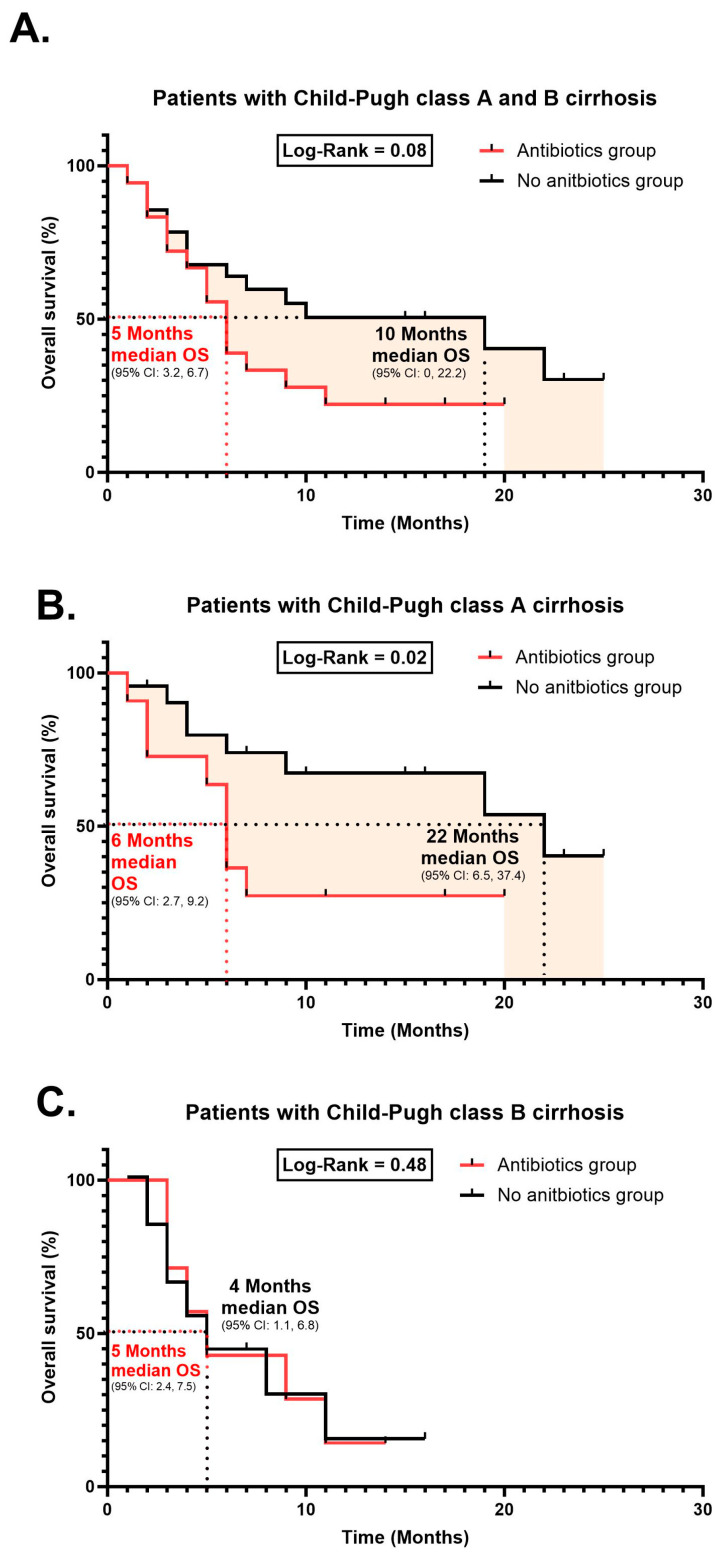
Kaplan–Meier survival curve of overall survival. (**A**) Patients with Child–Pugh class A and B cirrhosis. (**B**) Patients with Child–Pugh class A cirrhosis. (**C**) Patients with Child–Pugh class B cirrhosis.

**Table 1 cancers-16-00133-t001:** Baseline characteristics based on antibiotic use.

Variable	Total	No Antibiotics	Antibiotics	*p* Value
**Age (years)**	72 (65, 79)	70 (62, 78)	73 (67, 81)	0.23
**Male sex—n (%)**	51 (86.4)	35 (89.7)	16 (80)	0.30
**Body mass index BMI (kg/m^2^)**	25.8 ± 5.3	26.5 ± 5.4	24.5 ± 4.9	0.32
**ECOG performance score—n (%)**				
0–1	42 (71.2)	26 (66.7)	16 (80)	0.28
2–3	17 (28.8)	13 (33.3)	4 (20)	
**Child-Pugh class—n (%)**				
A	38 (64.4)	25 (64.1)	13 (65)	0.94
B	21 (35.6)	14 (35.9)	7 (35)	
**BCLC stage—n (%)**				
B	17 (28.8)	11 (28.2)	6 (30)	0.88
C	42 (71.2)	28 (71.8)	14 (70)	
**α-feto protein level—n (%)**				
Normal	20 (33.9)	16 (41)	4 (20)	0.10
Abnormal	39 (66.1)	23 (59)	16 (80)	
**Etiology of cirrhosis—n (%) ***				
Hepatitis B	16 (27.1)	12 (30)	4 (20)	0.25
Hepatitis C	12 (20.3)	6 (15.4)	6 (30)	
Non-viral	30 (50)	21 (53.8)	9 (45)	
**Comorbidities—n (%)**				
Cardiac disease	11 (18.6)	6 (15.4)	5 (25)	0.13
Diabetes mellitus	32 (54.2)	22 (56.4)	10 (50)	0.64
Chronic kidney disease	5 (8.5)	3 (7.7)	2 (10)	0.76
Hypertension	26 (44.1)	15 (38.5)	11 (55)	0.22
Cirrhosis	50 (84.7)	34 (87.2)	16 (80)	0.46

Data presented as n (%), mean ± standard deviation, or median (interquartile range) as appropriate. * One patient has no available etiology.

**Table 2 cancers-16-00133-t002:** Subgroup analysis of antibiotic types and the reason for use.

Antibiotic Types	Reason of Using	N (%)
Ceftriaxone	Spontaneous Bacterial Peritonitis (SBP)	6 (30%)
	Urinary Tract Infection (UTI)	1 (5%)
Ciprofloxacin	UTI	5 (25%)
Piperacillin/tazobactam	Sepsis	3 (15%)
Meropenem		1 (5%)
Augmentin	Upper Respiratory Tract Infections (URTIs)	2 (10%)
Moxifloxacin	Pneumonia	2 (10%)
**Total**		20 (100%)

**Table 3 cancers-16-00133-t003:** Treatment intervention categorized on the basis of antibiotics usage.

Variable	Total	No Antibiotics (*N* = 39)	Antibiotics (*N* = 20)	*p* Value
**Nivolumab Therapy—n (%)**				
First line	8 (13.6)	6 (15.4)	2 (10)	0.12
Second line	49 (83.1)	33 (84.6)	16 (80)	
Third line	2 (3.4)	0	2 (10)	
**Number of cycles**	7	7	6.5	
**Beta blockers—n (%)**	19 (32.2)	11 (28.2)	8 (40)	0.39
**Oral hypoglycemics—n (%)**	11 (18.6)	9 (23.1)	2 (10)	0.22
**Proton pump inhibitors—n (%)**	11 (18.6)	7 (17.9)	4 (20)	0.84
**Steroids—n (%)**	20 (33.9)	14 (35.9)	6 (30)	0.65

**Table 4 cancers-16-00133-t004:** Treatment response.

Variable	Total	No antibiotics	Antibiotics	*p* Value
**Response Based on mRECIST, n (%)**				
**CR**	1 (2.4%)	1 (3.7%)	0	0.87
**PR**	17 (41.5%)	11 (40.7%)	6 (42.9%)	
**SD**	16 (39%)	10 (37%)	6 (42.9%)	
**PD**	7 (17.1%)	5 (18.5%)	2 (14.3%)	
**Mean follow-up period**	**7.03 months**			

Complete response (CR), partial response (PR), progressed disease (PD), and stable disease (SD).

**Table 5 cancers-16-00133-t005:** Multivariate analysis of mortality.

Dependent Variable: Time to Death or Last Contact	RR	95% CI		*p* Value
**Antibiotic use**	3.30	1.04	10.39	0.04
**Gender**	0.62	0.14	2.63	0.51
**Age**	1.01	0.94	1.09	0.61
**Diabetes mellitus**	1.02	0.34	3.06	0.96
**Hypertension**	0.93	0.24	3.57	0.91
**Cardiac disease**	0.60	0.12	3.04	0.54
**Chronic kidney disease**	0.87	0.08	9.04	0.91
**Presence of cirrhosis**	1.40	0.38	5.05	0.60

## Data Availability

The datasets generated and/or analyzed during the current study are available from the corresponding author upon reasonable request and after signing a sharing agreement in accordance with the policies of KAIMRC for 3 years after the publication of this paper.

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
