# Peer review of "Antibiotic Exposure Concurrently with Anti-PD1 Blockade Therapy Reduces Overall Survival in Patients with Child–Pugh Class A Advanced Hepatocellular Carcinoma"

_cancers, 2023, doi:10.3390/cancers16010133_

Round 1
Reviewer 1 Report
Comments and Suggestions for Authors
The current study retrospectively analyzed whether the use of antibiotics during anti-PD1 blockade therapy affected the overall survival in HCC patients. Although the data presented are clinically meaningful and accords with preceding studies, logical explanations for the results are lacking.
Journal: Cancers
Title: Antibiotic exposure concurrently with anti-PD1 blockade therapy reduces overall survival in patients with Child-Pugh class A advanced Hepatocellular Carcinoma
Type: Retrospective cohort study
Main Question:
Does administrating of antibiotics during anti-PD1 blockade therapy affect overall survival in HCC patients?
Review on 2023.11.20
The current study retrospectively analyzed whether the use of antibiotics during anti-PD1 blockade therapy affected the overall survival in HCC patients. Although the data presented are clinically meaningful and accords with preceding studies, logical explanations for the results are lacking, detailed as follows:
Major issue
1. The authors presented the use of antibiotics during ICI treatment reduced overall survival in Child-Pugh class A HCC. However, subgroup analyses of intervention (antibiotics) are lacking – including the duration of use, types, time of administration (before / after Nivolumab injection) which are crucial for the main argument.
2. It should be more noted that the use of antibiotics could have been a proximate cause, while the root cause was infection, given that bacterial infection significantly affects the prognosis of patients with cancer.
3. The authors argued the disturbunce of microbiota by antibiotics was the underlyng cause of reduced overall survival in patients with Child-Pugh class A. However, there was no differences of overall survival in patients with Child-Pugh class B, and there is no explantion for this in the article. Does admistration of antibiotics not affect the micribiota in patients with Child-Pugh class B as much in class A? Or does it disturb the microbiota but do not affect the overall survival?
4. The mRECIST had to be assessed regarding the Child-Pugh class, given that there was significant discrepancy of overall survival by antibiotics in Child-Pugh class A and B. Furthermore, the authors must present the follow-up period for mRECIST, since the mRECIST assessment usually change over time (e.g. PR -> SD -> PD) in most patients with HCC.
5. The limitations and explanations of the study described by authors are mostly hard to understand.
A. In discussing the different results in overall survival and mRECIST assessment, the authors aruge that the use of antibiotics may not affect the immediate response to ICI treatment but affect in the long-term survival. But they do not present any evidence for their argument. Considering the use of antibiotics only within 30 days from the injection of Nivolumab were included in this study, it is hard to understand that the use of antibiotics affected in long term but not in the short term.
B. In discussion, the authors mention that there were some fluke data in patients with BCLC stage B, but they did not present those in the main text.
C. The authors mention including one-third of patients with Child-Pugh class B was one of their limitations, which is not understandable. Not including patients with Child-Pugh class C seems one of their limitations, regarding the small number of patients included.
D. The authors mention not recording the ‘routine’ locoregional therapy was one of the limitations, but it is of low possibility that it affected the results if it was ‘routine’ therapy that most patients had.
Minor issue
1. English grammer and mis-leading words must be corrected.
A. Please unify the term indicating the treatment and comparison group such as “group with/without antibiotics”.
B. In 3.3., correct the sentence “Notably, patients’ survival curves … was a significantly decrease in median overall survival…”
C. In 3.3., change the term ‘achived’ in the sentence “The no antibiotics use group achived 18.5% PD compared to 14.3% in the antibiotics use group.”
D. In 3.3, correct the term ‘disease stability’ to ‘SD’ and check the grammar in the sentence “With regards to disease stability, both groups the no antibiotics use groups and antibiotics use group achived similar results…”
E. In 3.3, delete the sentence “although this was not significantly different” after “In the mRECIST (Table 3), only one patient achieved CR in the antibiotic use comparted to non in the no antibiotic use”.
F. In discusssion, correct the grammar of the sentence “In addition to explore potential strategies to relieve the negative effects of antibiotics on ICI treatment efficacy”.
2. In 2.1, the authors must explain why they limited study population as ‘patients with advanced HCC and Child-Pugh classes A and B who received Nivolumab as a systemic therapy’.
3. In 3.2, the authors must explain why ‘beta blockers’, ‘oral hypoglycemics’, ‘proton pump inhibitors’, and ‘steroids’ are notable variables.
4. In 3.3, the information that ‘Child-Pugh class A is the recommended class to receive ICIs by the FDA’ should rather be mentioned in introduction or discussion.
5. In Table 3., the positions of PD and SD should be changed, considering their meanings.
6. In discusion the senetence “Furthermore, our study did not find any significant differences in the partial or complete response rates to ICIs between patients who recived antibiotics and those who did not according to mRECIST disease response criteria” should go backward and rather be described with other limitations.
Comments on the Quality of English Language.
Reviewer 2 Report
Comments and Suggestions for Authors
This study evaluated the impact of antibiotic use on the clinical outcomes of patients with hepatocellular carcinoma (HCC) undergoing treatment with immune checkpoint inhibitors (ICIs). The results showed that the concurrent use of antibiotics during treatment with ICI can have a significant detrimental impact on the overall survival in patients with HCC and Child-Pugh class A cirrhosis, but have no significant difference in the objective response rate between patients who received antibiotics and those who did not. There are some concerns as listed in the following:
(1) Give some information about that in what clinical situations the antibiotics were prescribed and what kinds of antibiotics were used for those HCC patients.
(2) L47: However, the data showed no significant difference in the objective response rate between patients who received antibiotics and those who did not. These findings suggest that antibiotic use may negatively affect survival outcomes in HCC patients undergoing ICIs treatment, potentially due to antibiotic-induced alterations to the gut microbiome impacting the ICI response. > The latter conclusion (antibiotic-induced alterations to the gut microbiome impacting the ICI response) seems to be unmatched with the data presented (the data showed no significant difference in the objective response rate between patients who received antibiotics and those who did not).
(3) L192: Table 3: give explanation why the number of total patients is 41 (27+14), but not 59 (39+20)
(4) Typos and others:
*L52: the Keywords Nivolumab and anti-PD-1 did not appear in the Abstract.
**L72-73: give brief introduction of PD-1 and CTLA-4
L137: (Table 1) shows
*L143: Table 1: Etiology of cirrhosis - n (%)
Hepatitis B 16 (27.1) +Hepatitis C 12 (20.3)+ Non-viral 30 (50) =58 <59
L182: patients’ survival curves -> Patients’ survival curves
L188: in (Table 3). -> in Table 3.
**L192: Table 3: 1 (3.7%) -> 1 (7.1%=1/14)
*L196: PR, 40% vs 42.9%, respectively. -> PR, 40.7% vs 42.9%, respectively.
*L211: our study? along with others [25-29] suggest that the use of antibiotics concurrently with ICI could contribute to reduced responses to treatment. -> but the present study showed no significant difference on treatment response (Table 3).
L240: outcomes. In addition to explore potential strategies to relieve the negative effects of antibiotics on ICI treatment efficacy.
L281: References: inconsistent writing format for the Journal name: Ref 5, 7, 13, 14, 15, 18, 21, 23, 24, 26, 27, 28
Comments on the Quality of English LanguageMinor editing of English language required
Reviewer 3 Report
Comments and Suggestions for Authors
This is an interesting small study of the impact of antibiotics on responses of advanced hepatocellular carcinoma subjects to anti-PDL therapy. Despite numerous caveats, such as subjects requiring antibiotics having conditions that might suggest worse outcomes, the paper suggests that more research should be done with this possible connection.
Antibiotic exposure concurrently with anti-PD1 blockade therapy reduces overall survival in patients with Child-Pugh class A advanced Hepatocellular Carcinoma, addresses failure to respond to PDL1 therapy as possibly due to prior antibiotic treatment altering the gut microbiota.
I have reviewed other papers on this topic. However, it is clear more needs to be done to substantiate this notion and fill in specific details, particularly the link to anti-PDL1 therapeutic failure. This work broadens this type of investigation into this specific niche, supplying suggestive evidence.
The paper supplies information to support this claim with a long list of potentially confounding factors (the link between co-morbidities that prompt antibiotic treatment and generally worse outcomes consequently), still supporting the general thesis that upsetting the gut microbiota with antibiotics might predict PDL1 therapeutic failure, as a statistical argument.
I would suggest listing the specific antibiotics used in these subjects with an annotation of the predicted impact on general classes of microbiota. This might be revealing due to the limited number of centers providing the subjects and possibly limited types of antibiotics used in these cases. If the antibiotics cover the entire gamut of possibilities, it will seem to weaken the thesis of the paper, since much of the work on this topic suggest that specific classes of microbiota would be responsible.
Of course, sampling stool microbiota and screening for microbial genus level alterations that correlate with anti-PDL1 responses would greatly strengthen the claim and provide independent support even if the antibiotics used here were of broad classes.
I suggest that the paragraph starting at line 245 be labeled with the header, 5. Limitations.
The English seems fine, with 'careful' misspelled on line 35.
Comments on the Quality of English LanguageOK
Round 2
Reviewer 1 Report
Comments and Suggestions for Authors
The present manuscript form has been well revised regarding the reviewers' opinion. However, there are some remaining issues to be addressed.
1. The main argument of this study seems to be 'the significant differences of overall survival in the study patients were likely due to the use of antibiotics which lowered the efficacy of ICI treatment by causing the disruption of gut microbiota.' However, the data presented in the initial manuscript was not sufficient to prove that hypothesis, which was the reason additional subgroup analysis was asked for. It would be more convincing if the data in the subgroup analysis back up the main argument (e.g. the patients treated antibiotics for longer duration had lower survival rate). Additional analysis including whether the duration of administration, mode of administration (e.g. oral, intravenous), therapeutic class (e.g. beta lactams, fluoroquinolones, tetracyclines) of antibiotics affected the overall survival are recommended.
2. Corrections of the English grammar are recommended including the lines below:
L45-46, L166-168, L169-171, L176
3. In the revised manuscript, there are still some expressions such as ‘groups without antibiotics’, 'no antibiotic group', 'cohort with no antibiotic use'. Unifying the terms for the group is recommended.
Comments on the Quality of English LanguageMinor editing of English language required.
Author Response
The present manuscript form has been well revised regarding the reviewers' opinion. However, there are some remaining issues to be addressed.
- The main argument of this study seems to be 'the significant differences of overall survival in the study patients were likely due to the use of antibiotics which lowered the efficacy of ICI treatment by causing the disruption of gut microbiota.' However, the data presented in the initial manuscript was not sufficient to prove that hypothesis, which was the reason additional subgroup analysis was asked for. It would be more convincing if the data in the subgroup analysis back up the main argument (e.g. the patients treated antibiotics for longer duration had lower survival rate). Additional analysis including whether the duration of administration, mode of administration (e.g. oral, intravenous), therapeutic class (e.g. beta lactams, fluoroquinolones, tetracyclines) of antibiotics affected the overall survival are recommended.
Thank you for your insightful comments, we highly appreciate the time and effort you have put toward reviewing our work.
We would like to provide some clarification regarding your suggestion for subgroup analysis based on the duration of antibiotic administration, mode of administration (e.g., oral, intravenous), and therapeutic class (e.g., beta lactams, fluoroquinolones, tetracyclines).
Our subgroup involves only 20 patients and most of them had antibiotic for a duration of 5-8 days, and only 4 patients received antibiotic treatment for 14 days either orally or intravenously.
When we did a preliminary analysis, we found no significant difference between the patients treated for 5-8 days and patients treated for 14 days. This could be due to the limited sample size and the narrow range of treatment durations.
We understand and appreciate your concern to strength our main hypothesis, therefore we have included a table that list the names of the antibiotic used along with reference (Inserted in the text) from the literature which shows their potential role in the disruption of gut microbiota (all the antibiotic used have shown a significant role in disturbing the gut microbiota). We hope that our data backed by the literature is sufficient to provide a substantial basis for the role of antibiotic use to lower the efficacy of ICI treatment.
- Corrections of the English grammar are recommended including the lines below:
L45-46, L166-168, L169-171, L176
Fixed
- In the revised manuscript, there are still some expressions such as ‘groups without antibiotics’, 'no antibiotic group', 'cohort with no antibiotic use'. Unifying the terms for the group is recommended.
Fixed
Thank you once again for your valuable suggestion and input.
Reviewer 2 Report
Comments and Suggestions for Authors
The present manuscript has revised a lot according to the Reviewers’ suggestions and comments. There are some concerns (typos and others) as listed in the following:
*L58: It is better to include the keyword Nivolumab into the Abstract.
*L83: can inhibits T cell regulates and activation
*L129: umol/L-> mmol/L
*L183: in (Table 2)-> in Table 2
*L194: in (Table 3) -> in Table 3
*L198: (p=12)-> (p=0.12)
*L203: (P >0.5 in all) -> (P >0.05 in all)
*L228-232: CR, PR, PD, SD -> full name first
**L229: Similar number of patients -> Similar percentage number of patients
L232: both groups the group…
L256: in table 2 -> in Table 2
L261: due to changes in the gut microbiota due to the antibiotic uses
L306: Lastly, Our relative
**L436: References: inconsistent writing format for the Journal name: Ref 5, 7, 13, 14, 15, 18, 21, 23, 24, 25, 26, 27, 28, 30, 31, 32, 33, 34, 35
5: Current Oncology -> Current oncology
7: Frontiers in Immunology -> Frontiers in immunology
13: Annual Review of Medicine
14: New England Journal of Medicine
15: Future Oncology
18: Clinical Cancer Research
21: Digestive Diseases
23: Journal of Hematology & Oncology
24: Frontiers in Immunology
25: Journal of Thoracic Oncology,
26: Gynecologic Oncology
27: Journal of Clinical Medicine
28: Liver Cancer
30: Infection and Immunity
31: Journal of Antimicrobial Chemotherapy
32: Clinical Infectious Diseases
33: European Journal of Clinical Microbiology and Infectious Diseases
34: PLoS One
**35: Antimicrobial Agents and Chemotherapy, 2019. 63(10): p. 10.1128/aac. 00820-19. [2019 Sep 23;63(10):e00820-19. doi: 10.1128/AAC.00820-19. Print 2019 Oct.]
Comments on the Quality of English LanguageModerate editing of English language required
Author Response
The present manuscript has revised a lot according to the Reviewers’ suggestions and comments. There are some concerns (typos and others) as listed in the following:
*L58: It is better to include the keyword Nivolumab into the Abstract.
Fixed
*L83: can inhibits T cell regulates and activation
Fixed
*L129: umol/L-> mmol/L
Thank you for your suggestion, we would like to keep the µmol/L as this is the unit that was used to report the number.
*L183: in (Table 2)-> in Table 2
Fixed
*L194: in (Table 3) -> in Table 3
Fixed
*L198: (p=12)-> (p=0.12)
Fixed
*L203: (P >0.5 in all) -> (P >0.05 in all)
Fixed
*L228-232: CR, PR, PD, SD -> full name first
Fixed
**L229: Similar number of patients -> Similar percentage number of patients
Fixed
L232: both groups the group…
Fixed
L256: in table 2 -> in Table 2
Fixed
L261: due to changes in the gut microbiota due to the antibiotic uses
Fixed
L306: Lastly, Our relative
Fixed
**L436: References: inconsistent writing format for the Journal name: Ref 5, 7, 13, 14, 15, 18, 21, 23, 24, 25, 26, 27, 28, 30, 31, 32, 33, 34, 35
5: Current Oncology -> Current oncology
7: Frontiers in Immunology -> Frontiers in immunology
13: Annual Review of Medicine
14: New England Journal of Medicine
15: Future Oncology
18: Clinical Cancer Research
21: Digestive Diseases
23: Journal of Hematology & Oncology
24: Frontiers in Immunology
25: Journal of Thoracic Oncology,
26: Gynecologic Oncology
27: Journal of Clinical Medicine
28: Liver Cancer
30: Infection and Immunity
31: Journal of Antimicrobial Chemotherapy
32: Clinical Infectious Diseases
33: European Journal of Clinical Microbiology and Infectious Diseases
34: PLoS One
**35: Antimicrobial Agents and Chemotherapy,
Thank you, all fixed, please note that in the word file the reference management changes from computer to another, therefore we have provided a PDF copy for the final version.
Thank you for your insightful comments and suggestions, we highly appreciate the time and effort you have put toward reviewing our manuscript.